# Glycoproteins Presenting Galactose and *N*-Acetylgalactosamine in Human Seminal Plasma as Potential Players Involved in Immune Modulation in the Fertilization Process

**DOI:** 10.3390/ijms22147331

**Published:** 2021-07-08

**Authors:** Justyna Szczykutowicz, Joanna Tkaczuk-Włach, Mirosława Ferens-Sieczkowska

**Affiliations:** 1Department of Biochemistry and Immunochemistry, Wroclaw Medical University, 50-369 Wrocław, Poland; justyna.szczykutowicz@umed.wroc.pl; 2Laboratory of Diagnostic Techniques, Medical University of Lublin, 20-081 Lublin, Poland; joannatkaczukwlach@umlub.pl; 3Family Health Centre AB OVO, 20-819 Lublin, Poland

**Keywords:** infertility, glycome, T/Tn antigens, carbohydrate-protein interaction, maternal tolerance

## Abstract

In light of recent research, there is increasing evidence showing that extracellular semen components have a significant impact on the immune reaction of the female partner, leading to the tolerogenic response enabling the embryo development and implantation as well as further progress of healthy pregnancy. Seminal plasma glycoproteins are rich in the unique immunomodulatory glycoepitopes that may serve as ligands for endogenous lectins that decorate the surface of immune cells. Such interaction may be involved in modulation of the maternal immune response. Among immunomodulatory glycans, Lewis type antigens have been of interest for at least two decades, while the importance of T/Tn antigens and related structures is still far from understanding. In the current work, we applied two plant lectins capable of distinguishing glycoepitopes with terminal GalNAc and Gal to identify glycoproteins that are their efficient carriers. By means of lectin blotting and lectin affinity chromatography followed by LC-MS, we identified lactotransferrin, prolactin inducible protein as well as fibronectin and semenogelins 1 and 2 as lectin-reactive. Net-O-glycosylation analysis results indicated that the latter three may actually carry T and/or Tn antigens, while in the case of prolactin inducible protein and lactotransferrin LacdiNAc and lactosamine glycoepitopes were more probable. STRING bioinformatics analysis linked the identified glycoproteins in the close network, indicating their involvement in immune (partially innate) processes. Overall, our research revealed potential seminal plasma ligands for endogenous Gal/GalNAc specific lectins with a possible role in modulation of maternal immune response during fertilization.

## 1. Introduction

Over the last few decades, scientists have been trying to explain the intriguing mystery in immunology, which is keeping male gametes and a fetus, presenting non-self-paternal antigens, safe from maternal immune reaction. Various hypotheses have been proposed to explain this phenomenon, including fetal antigen immaturity, strong suppression of the mother’s immune system during pregnancy, and strict impermeability of the blood-placenta barrier. None of these hypotheses was fully confirmed. Some recent data indicate the engagement of seminal plasma (SP) components in the interactions involved in the creation of an appropriate maternal immune balance, allowing selective tolerance for gametes and developing trophoblast, without inactivation of the effective response to the infections [1].

According to the fetoembryonic defense hypothesis, the interactions involved in the development of immune tolerance during fertilization and pregnancy are based on recognition of the sugar moieties by the maternal immune system [2,3,4]. The hypothesis assumes that these protein–carbohydrate bindings in the female reproductive tract in the post-coital period are similar to the mechanism exploited by pathogens and metastatic cancer cells to escape from the control of host immune system [3,5]. Both pathogens and cancer cells present surface glycan antigens that are normally absent (or extremely rare) in the glycoproteins of healthy human body fluids. Surprisingly, similar carbohydrates are abundant on the sperm surface and in seminal plasma glycoproteins of healthy, fertile men [4,6]. So far, the research related to the analysis of semen glycosylation pattern has been focused on the glycoepitopes containing fucose and sialic acid, considered to be immunomodulatory [7,8,9,10,11,12]. Such terminal glycoepitopes can be presented on both N- and O-glycans. To date, much more research concerns the N-glycome because of its relative technical ease. This approach, however, removes from the field of interest an important category of truncated O-glycans, including T and sialo T/Tn antigens, though they are also assigned with an immunomodulatory role (Figure 1) [13,14,15]. 

Thomsen–Friedenreich related antigens, T (Galβ1-3GalNAcα1-R), Tn (T precursor, GalNAcα1-R) and sialyl-Tn (NeuAcα2-6GalNAcα1-R) are known as cancer associated antigens. Such antigens are present in approximately 90% of all cancers. Though T-antigen may be also expressed in the normal epithelium, Tn seems to be strictly cancer-related [16,17,18,19,20]. Thomsen–Friedenreich related antigens have been correlated with cancer progression and poor prognosis [13,21,22,23,24]. Interestingly, it has been also suggested that predominance of Tn over T is associated with a high metastatic potential of carcinoma, thus resulting in more effective evasion from the immune system control [17,18,19]. 

Mucin-type glycosylation was confirmed in seminal plasma components by Milutinović et al. [25] and different O-glycan structures, including T antigen, were earlier reported by Pang et al. [4]. There is no information to show whether T/Tn antigens may be the next category of glycospecies important for immunomodulation during fertilization. In the current work, the two plant lectins, *Vicia villosa* (VVL) and *Maclura pomifera* (MPL), were used to detect truncated O-glycans and related structures in the seminal plasma glycoproteins. VVL preferentially binds a single N-acetylgalactosamine residue attached directly to serine or threonine in the polypeptide backbone (Tn antigen) [26,27,28]. The lectin may interact also with GalNac-Gal disaccharides. Galβ1-3GalNac (T antigen) is favored by the MPL, which has the ability to bind the sialylated forms (sialo-T and -T antigen) as well as those without the terminal sialic acid residue [27,29,30].

The aim of the study was to identify glycoproteins that can act as carriers of the above mentioned structures in seminal plasma, and to examine the expression of these antigens among SP glycoproteins. Our research revealed lactotransferrin, prolactin inducible protein, fibronectin and semenogelins 1 and 2 as proteins presenting glycans terminated with Gal or GalNAc residues. We also suggest the domination of T over Tn antigen in some glycoproteins in non-fertile subjects compared to the control group. Bioinformatics analysis assigned the identified glycoproteins as involved in immune processes, pointing out their possible role in modulating the mother’s immune response during fertilization.

## 2. Results

### 2.1. Detection of MPL and VVL Ligands in Seminal Plasma

Lectin ligands were detected in seminal plasma glycoproteins of the men with different semen patterns. The samples were separated by SDS-PAGE and then transferred onto nitrocellulose membrane. The blots prepared in this way were probed with the lectins specific for truncated O-glycans and related structures: VVL (Tn antigen) and MPL (T antigen). The patterns of SP reactivity with the lectins are presented in Figure 2A and Figure 3A. In all the analyzed samples, glycoproteins assigned with 83 kDa, 70 kDa and 32 kDa molecular masses bound the VVL lectin. The reactivity also appeared in the 135 kDa and 15 kDa bands, though the bands were stained weakly in many samples, with no correlation with the diagnostic group to which the sample has been classified. In all the samples, reactivity with VVL was also observed in the short glycopeptides (<15 kDa) resulting from proteolytic cleavage. More signals were observed when analyzing the MPL reactivity of SP. Furthermore, glycoprotein fractions of 83, 70 and 32 kDa dominated in this case, but additional bands of 135 and 53 kDa were observed in all the samples. In contrast, the 15 kDa band was not visible in most of the samples, while glycopeptides <15 kDa also reacted with MPL lectin.

### 2.2. Evaluation of the Expression of T and Tn Glycoepitopes in Individual Seminal Plasma Samples

Selected glycoprotein bands were subjected to densitometric analysis in order to assess, on the one hand, the content of the analyzed glycoepitopes measured as total lectin reactivity (Figure 2B and Figure 3B) and, on the other hand, density of particular glycoepitope on the respective carrier proteins defined as the ratio of total lectin reactivity to protein content in bands (Figure 2C and Figure 3C). When the expression of Tn antigens was examined by means of its reactivity with VVL, the wide dispersion of individual values was observed, especially in the oligozoospermic group in the 83 kDa band (Figure 2B). Interestingly, this spread was not so large when glycoepitopes expression was analyzed in relation to the protein content (Figure 2C). This may indicate that the fluctuations in individual semen samples are mainly related to the expression of the protein itself, while maintaining an unchanged glycosylation profile. Comparing VVL reactivity in the individual patients’ groups, statistically significant differences were observed in glycopeptide fraction <15 kDa, both in terms of total lectin reactivity and protein glycoepitope density in the oligoasthenozoospermic group when compared to the control subjects (Figure 2B,C). Such a decrease in the density of Tn antigens in glycopeptides <15 kDa was also observed in the group of normozoospermic men with reduced fertility. Analysis of MPL reactivity, considered as a measure of the T antigen content, showed differences versus controls in overall reactivity with the lectin in the 83 kDa and 70 kDa bands. In the pathological groups, the reactivity was increased comparing to the control group in A and OA subjects in gp 83 kDa, and in N and O groups in gp 70 kDa (Figure 3B). Gp 83 kDa showed the increased MPL reactivity also when compared OA to the group of infertile men with normal semen pattern (N). As in the case of VVL, also for MPL-reactive glycoproteins, the observed differences were related to the expression of a particular protein itself rather than to a change in the density of lectin-reactive glycoepitopes on it (Figure 3C). Statistical significance was demonstrated only for the difference in glycoepitope density between N and OA groups in the 83 kDa band. The difference related to the increase in reactivity in the 32 kDa band was observed in group A in relation to N-normozoospermic infertile patients. Interestingly, an opposite pattern can be observed when comparing the VVL- to MPL- reactivity in the 70 kDa band. It may be related with opposite tendency to present Tn and T antigens in this fraction. Determination of VVL/MPL- reactivity ratio showed significantly lowered content of Tn relative to T antigen in pathological groups especially in N, A and OA compared to control group (Figure 3D).

### 2.3. Identification of Proteins in Lectin-Reactive Bands

Lectin-reactive protein fractions were excised from the gel after SDS-PAGE and CBB (Coomassie Brilliant Blue) staining, then trypsin-digested and analyzed by means of LC-MS to identify the proteins. Table 1 presents the results of the protein content examination of the 135-, 83-, 70-, 53-, 32-, 15- and <15-kDa fractions. Only records with scores exceeding 1000 were considered. The detailed report is available in the Appendix A (see Appendix A). In the 135 kDa fraction, the dominant matches represented mucin 6 (MUC6), aminopeptidase N (ANPEP), fibronectin (FN1) and lactotransferrin (LTF). The same proteins, accompanied with macroglobulin, appeared in the 83 kDa fraction (for ANPEP the score was <1000, see Appendix A). The 70 kDa fraction contained galectin-3 binding protein (LGALS3BP) as well as FN1 fragments and LTF. Fragments of MUC6 and clusterin (CLU) appeared with lower score values (Appendix A). In the further fractions, apart from the dominant proteins, more proteolytic fragments appeared. FN1, prostate specific antigen (KLK3), annexin, CLU and prostatic acid phosphatase (ACP3) dominated in the 32 kDa fraction. Zinc-alpha-2-glycoprotein (AZGP1), LGALS3BP, MUC6 and glycodelin (PAEP) appeared with lower scores. PIP dominated in the 15 kDa fraction, fragments of ACP3, FN1, CLU and both semenogelins were also present. Similarly, PIP was dominant in <15 kDa glycopeptides but also semenogelin 1 and 2 (SEMG1 and 2) fragments were abundant in these fractions.

### 2.4. Isolation of Lectin-Reactive Glycoproteins

In SDS-PAGE, electrophoretic fractions may contain a number of proteins, including those that do not react with the probe, which is confirmed by the presence of albumin (which is not a glycoprotein) among the MS records showed in Table 1. To confirm the identification of glycoproteins presenting the analyzed glycoepitopes, affinity chromatography on the spin-columns filled with immobilized lectins was applied for the isolation of VVL and MPL ligands from pooled normozoospermic semen. The eluates were neutralized, purified from the eluting sugars and concentrated. The electrophoretic images of the obtained isolates are presented in Figure 4A, B (lane 3 and 4). The presence of 83 kDa, 70 kDa, 53 kDa and 15 kDa glycoprotein fractions and smaller glycopeptides was thus confirmed as VVL and MPL ligands. 135 kDa fraction was present only in MPL isolate. LC-MS analysis of the obtained isolates (Table 2 and Appendix A) confirmed the presence of SEMG1 and 2, LTF, PIP, FN1 and KLK3 in both isolates (the latter two with much lower score for VVL isolate), and in the case of MPL-reactive glycoproteins additionally ANPEP and ACP3. The presence of dominant proteins was also confirmed by means of immunoblotting with appropriate antibodies (Figure 4A,B, lanes 5–9).

### 2.5. Bioinformatics Analysis

Glycoproteins identified as lectin-reactive were analyzed for the presence of potential O-glycosylation sites using the NetOGlyc platform (Figure 5). A score of >0.5 makes the Ser/Thr residues in the polypeptide chain likely to be attachment sites for GalNAc and subsequent sugars. As predicted, MUC6 had the largest number of O-glycosylation sites, in tightly packed clusters in the VNTR (variable number tandem repeat) region (amino acid residues ~1200–2300). The significant content of O-glycosylation sites was also confirmed for SEMG1 and 2 and FN1, although they were more dispersed in these proteins. The analysis practically ruled out the possibility of PIP O-glycosylation. It also showed that there were only a few potential O-glycosylation sites in LTF. These results indicate that in the case of latter glycoproteins, instead of T/Tn antigens some alternative glycospecies of N-glycans terminated with Gal/GalNAc should be considered as lectin ligands. 

STRING analysis was performed to look at the networks of potential interactions among the identified VVL- and MPL-reactive glycoproteins. Figure 6A presents the protein network when glycoproteins identified in SDS-PAGE bands were submitted for the analysis. The glycoproteins dominant in the bands with confirmed differences in the lectin reactivity in various patient groups were considered: SEMG1, SEMG2, MUC6, FN1, LGALS3BP1, PIP, LTF and α2-macroglobulin (LRP1) transferrin (TF). Figure 6B presents the analysis performed for the glycoproteins dominant in the lectin-chromatography isolates: the mentioned above set of proteins without LGALS3BP1, LRP1 and TF but completed with ACP3 and ANPEP. The network of functional linkages, established in part from experimentally confirmed interactions, and from protein co-expression and text-mining, bound—in both cases—both semenogelins, LTF and PIP. Subsequent junctions, via AZGP1 as a linker, lead to closely related integrin proteins, proto-oncogene tyrosine-protein kinase Src (SRC) and LRP1. Fibronectin also belonged to this part of the network. MUC6 found itself outside the network in both cases. Searching the database for the potential involvement of the analyzed proteins in biological processes/reactome pathways assigned the processes related to the immune response, including the innate one, for most of them (Figure 6C). Reactome pathway analysis linked the identified proteins with signaling processes, both integrin-dependent and related to MAPK, L1, PECAM and syndecans.

## 3. Discussion

Recent years have brought an increasing interest to the immunomodulatory role of protein–carbohydrate interactions. Numerous lectins that reside on the surface of immune cells and are able to recognize particular sugar sequences have been described so far. This category includes C-type lectin receptors, such as DC-SIGN (Dendritic Cell-Specific Intercellular adhesion molecule-3-Grabbing Non-integrin), MBP (Mannose Binding Protein), selectins, MGL (Macrophage Galactose-type Lectin) and others [31,32,33]. The other groups comprise siglecs, belonging to the immunoglobulin superfamily and galectins, respectively recognizing sialic acid and β-galactose [34,35,36,37]. 

The importance of the interaction of lectins with glycans is most often investigated while monitoring the interaction of pathogens with the host tissues or studying the mechanisms of tumor metastasis. In both of these—distant after all—processes the invasive cells try to “trick” the host’s immune system, suppressing its defensive response, redirecting it to the tolerogenic pathway and thus escaping from the immune control [34,35,37,38,39,40,41,42]. 

A health-threatening phenomenon in tumor metastasis will be a desired one in the fertilization process. Redirecting the immune response to the tolerogenic pathway involves tolerogenic dendritic cells [43,44,45]. This may prevent the attack on the spermatozoa and developing embryo. It has been reported that semen promotes differentiation of Tregs [46], which are necessary for the proper implantation of the trophoblast [47]. However, the direct mechanism of the immune response switch to the tolerogenic pathway remains speculative. 

Among the glycoepitopes considered to be immunomodulatory, apart from tri- and tetrasaccharides containing fucose and sialic acid (Lewis antigens), some truncated O-glycans are also being mentioned, including Thomsen-Friedenreich (T) and Tn antigens as well as their sialylated versions (sT and sTn). Their increased expression was observed in neoplastic cells, and the increased ratio of Tn/T epitopes was reported as associated with poor prognosis in neoplastic disease as long ago as at the end of the last century [16,17,18,19,48]; however, such studies have not lost their relevance until today [13,49,50,51,52]. O-glycosylation was also studied earlier in seminal plasma. Applying MALDI-MS, Pang et al. showed a set of SP O-glycans, including T-antigen structure [4]. Milutinović at al. analyzed mucin-type glycosylation, suggesting potential reactivity with DC-SIGN lectin annotated to O-linked Lewis-type antigens [25]. The expression of truncated O-glycans has not been studied in detail so far. 

Among endogenous lectins capable of recognizing galactose and its derivatives, the two groups deserve attention. Galectins recognize β-galactosides and great interest on their impact on tumor progression has arised in the recent years [37,42]. Macrophage galactose binding lectin (MGL), present on the surface of macrophages and the other cells of immune system, is specific for GalNAc and involved in regulation of the immune response [53,54,55,56,57,58].

In the current study, we chose the two plant lectins as tools to analyze the presentation of glycans terminated with Gal or GalNAc residues. VVL is reported to be specific for Tn antigen. MPL specificity has been described as GalNAc residue in the proximity of the peptide backbone, as Ser/Thr flanking amino acids may be bound by the lectin secondary binding site [59]. As elongation of the sugar chain with galactose does not change the binding, MPL is often reported as T-antigen specific [27,60,61]. Thus, the selected lectins enable the insight in glycans with the terminal, interactable residues of both monosaccharides of interest. Our team made some attempts to use these two lectins while looking for glycoepitopes that could be associated with male infertility [62,63]. We found that SP reacted with a number of O-specific lectins, including VVL and MPL. Considering even the somewhat overlapping lectin specificities, we concluded that expression of Gal/GalNAc presenting O-glycans was diminished in infertile compared to control subjects [63]. Decrease in the reactivity was the most prominent in O group [62]. In both studies unfractionated SP was analyzed. As proteomic analysis of seminal plasma suggested distinct protein synthesis pathways to be active, in particular, glands and cells of the male reproductive tract [64], such proteins may also present distinct glycosylation patterns. Thus, we felt prompted to provide more detailed analysis aimed at identification of glycoproteins that are efficient carriers of glycoepitopes with a terminal Gal or GalNAc residue. As shown in the current study, both MPL and VVL lectins reacted with the same dominant glycoprotein fractions. Some differences in reaction intensity were also observed, most probably related to the occurrence of T apart of Tn antigen. 

Mass spectrometry analysis revealed LTF and FN1 as the dominant proteins in VVL- and MPL-reactive 83 kDa and 70 kDa fractions. Both proteins were also released from the chromatographic columns filled with MPL- or VVL- agarose. Thus, LTF and FN1 can be considered as ligands for VVL and MPL lectins thereby carriers of glycans presenting terminal Gal/GalNAc residues. The increase in MPL reactivity in the pathological groups of subjects in LTF and FN containing bands was observed, especially for A and OA in 83 kDa and N and O in 70 kDa. However, when the total lectin reactivity was standardized against the protein content in each sample and each selected band, there were no alternations between control and pathological group of patients. These results suggest that the observed differences in MPL reactivity may be related to the expression of a particular protein itself rather than to a change in the density of lectin-reactive glycoepitopes on it. These results may support former data showing the increased expression of LTF in infertile men with oligo-, astheno- and oligoasthenozoospermia compared to normospermic ones [65,66]. 

No statistically significant differences were observed among the groups of patients in VVL reactivity for 83 kDa and 70 kDa bands. Carbohydrate specificities of both used lectins overlap to some extent. While VVL binds GalNAc (Tn antigen) and MPL, it is also able to recognize the same glycoepitope, preferring the extended version of T antigen. Thus, regarding lectins’ preferences, our result may be also read as the prevalence of T versus Tn in pathological groups (N, A, OA) over the control group. An opposite tendency to bind both lectins in fertile and infertile men can be observed in the 70 kDa band, which is also visualized in Figure 3D as the ratio of VVL/MPL reactivity. This observation may support the concept of more tolerogenic Tn properties. An increased ratio of Tn over T antigen is associated with poor prognosis and high metastatic potential in neoplastic diseases, thus in the context of conception, the advantage of Tn antigen over T in the seminal fluid may be related to more effective development of maternal tolerance towards male gametes and increased chance for trophoblast implantation. Moreover, it was demonstrated in the mouse muscle tissue model that knockdown of T-synthase, which is responsible for the formation of T antigen by attachment of galactose residue to truncated Tn antigen, suppresses the function of CD8+ T cells and enhances galectin-1 secretion [67]. This skews the immune response towards a Th2 cytokine profile, which enables the achievement of immunological tolerance toward skin allografts in mice. Similarly, immune deviation towards Th2/Treg response is associated with pregnancy and reproductive success. Thus, it seems possible that the dominance of truncated Tn over T antigen in seminal plasma may be important in the achievement of proper immune balance for successful fertilization and pregnancy. To sum up, our observation may indicate a possible regulatory role of LTF and/or FN1, as a carrier of immunomodulatory glycoepitopes, in generating maternal tolerance during the fertilization process. 

Regarding the O-glycosylation potential of mentioned proteins, the bioinformatic NetOGlyc analysis revealed a number of potential O-glycosylation sites in FN1, scattered through the polypeptide backbone. This protein is fragmented during seminal plasma liquefaction and may be a source of peptides that present T/Tn glycoepitopes in diverse ways. LTF contains some sites with NetOGlyc score >0.5, and therefore can be O-glycosylated, but they are significantly fewer than in the case of massive O-glycosylation of FN1 or SEMG1 and 2. It may be possible that LTF serves as a carrier only for one out of the two antigens of interest (T or Tn), thus the different pattern of reactivity with the two lectins may be due to the different ratio of the two LTF glycoforms.

Interestingly, the 15 kDa fraction, which mainly contains PIP, was intensively stained in Western blots probed with both MPL and VVL. It was also an abundant protein in the fractions released from chromatographic columns. The analysis of potential O-glycosylation sites virtually excludes the presence of such modification in PIP, which is consistent with the literature data confirming the presence of a single N-linked glycan in the protein structure. However, in this case it is worth considering the possibility of the presence of terminal, unmasked Gal/GalNAc residues in a particular configuration of N-glycans, in the form of LacNAc or LacdiNAc antigens. Oligo-LacNAc repeats have been found in PIP single N-glycan [68]. PIP has been indicated as a candidate for galectin-3 ligand in prostasomes [69], although not all the available studies confirm this particular interaction [70]. Involvement of PIP in immunoregulation was described in cancer and also suggested contribution to seminal plasma immunosuppresive properties [71,72]. Many recent reports point to the presence of an antigen called lacdiNAc (GalNAcβ1-4GlcNAc) in glycans of a KLK3 in the context of neoplastic transformation [73,74,75,76,77]. The occurrence of the above mentioned structures may be also considered in LTF. Reactivity of such glycans with endogenous lectins may mimic these of Tn and T antigens respectively. 

The fraction of increasing interest, also in our former studies [78], contained the glycopeptides of <15 kDa, composed mainly with SEMG1 and 2 fragments, resulted from KLK3-catalyzed cleavage of semen coagulum. In this fraction, differences were observed between the control group and the pathological normozoospermic group, indicating a higher content of the GalNAc antigen in the former one. SEMG1 and SEMG2 are predominant proteins of human seminal plasma, which participate in the formation of gel-like coagulum of newly ejaculated semen. Most of the available studies claim that while SEMG2 is a glycoprotein, SEMG1 is devoid of glycosylation. However, our results appear to contradict these reports. The analysis of glycosylation sites using the NetOGlyc platform showed that although there were fewer potential O-glycosylation sites in SEMG1 than in SEMG2, their number and score values strongly suggested the possibility of such modification. Moreover, SEMG1 was also identified in lectin column isolates, indicating the presence of lectin-recognized glycans. It is worth mentioning that the original study questioning the presence of glycosylation in SEMG1 dates back to 1996 [79] and applied the methodology available at that time, not very sensitive to today’s point of view. Recently, SEMG1, similarly to SEMG2, was found to be a potential Galectin-3 ligand, although in this case the authors, based on the above-mentioned reports, questioned the possible mechanism of SEMG1-Gal-3 binding [69]. Though co-isolation of SEMG1 via binding of the SEMG2 counterpart cannot be excluded, in our opinion, existence of some O-glycans in this protein is equally possible.

The participation of potential carriers of seminal plasma T/Tn antigens in the regulation of the immune response was confirmed by the STRING analysis, which showed 11 GO terms in the reactome network as involved in immune reaction process, 5 (Figure 6A) or 6 (Figure 6B), of which are involved in the innate immune response. Moreover, functional enrichment analysis pointed out MAPK signaling as one of the main pathways in the analyzed network. The MAP kinase cascades have the well-established role in cancer cell progression as well as in T cell development and their function. Together, it seems possible that the analyzed proteins, shown as VVL and MPL ligands, participate in immunoregulation through interaction with lectins of the immune system (MGL, Gal) affecting differentiation of DC and the Tregs population. Moreover, some proteins of interest are related to MAPK signaling, which may also support the regulation of T cell proliferation. Interestingly, the STRING analysis located MUC6 outside the network of the analyzed proteins. Furthermore, a relatively low score of MUC6 isolation with both lectins seems to be surprising, as this protein is the most likely carrier of O-linked T/Tn antigens [80,81,82], as indicated with the abundant, tightly packed O-glycosylation sites showed in NetOGlyc analysis as well as the bibliographic data. Typically, O-glycosylation is considered “mucin-like”. Meanwhile, our research shows that although we focus on classic O-glycans, it is not mucin that is the most effective carrier of glycoepitopes involved in regulatory interactions. It seems important to pay attention to these types of glycans in SEMG1, SEMG2 and FN1. 

## 4. Materials and Methods

### 4.1. Clinical Samples

The research material consisted of the samples of seminal plasma of men who visited the Family Health Centre AB OVO, Lublin, for standard examination of semen and further diagnosis of infertility. They were men aged 22–56, living in childless couples, despite unprotected sexual intercourse for at least a year. Detailed data regarding patients’ age are provided in Table 3. Diagnosed female infertility in the couple was an exclusion factor. All the female partners were characterized by no ovulation disorders and normal anatomy of the reproductive organ, confirmed by ultrasound examination. Semen samples were obtained by masturbation after a period of 2–5 days of sexual abstinence. After liquefaction of the semen and taking the material for standard semen analysis, the residue was centrifuged at room temperature (RT) for 10 min, 1500× *g*, in order to remove sperm without their licking. The supernatant was then centrifuged again at 10,000× *g*, 15 min, at 4 °C, to remove cellular and membrane debris. The obtained supernatant was protected from further proteolysis with an antiprotease cocktail, aliquoted and stored at −80 °C until used. The patients were grouped according to WHO directives as normozoospermic (n = 32), asthenozoospermic (n = 20), oligozoospermic (n = 15) and oligoasthenozoospermic (n = 20). The control group consisted of seminal plasma samples of volunteers with confirmed paternity within several recent years (n = 16). The participants gave their informed written consent for being enrolled in the study. The project has been approved by Wroclaw Medical University Bioethics Council (no. of approval: KB-23/2020). 

### 4.2. SDS-PAGE

Protein concentration in seminal plasma samples was determined using the bicinchoninic acid (BCA) protein assay following the manufacturer protocol (Thermo Fisher Scientific, Waltham, MA, USA). Sodium dodecyl sulphate polyacrylamide gel electrophoresis (SDS-PAGE) was conducted according to the standard Laemmli procedure [83] in a 12.5% gel. Two gels were electrophoresed simultaneously, one for SyproRuby staining (Sigma-Aldrich, St. Louis, MO, USA) and the other for protein transfer onto a nitrocellulose (NC) membrane in semi-dry blotter (Hoefer Inc., Holliston, MA, USA; 120 mA, 90 min). Constant protein amount was loaded on the gel lane, 4 µg for SyproRuby staining and 5 µg for lectin blotting. In order to standardize the evaluation of the protein in the analyzed protein bands, each gel contained a lane loaded with 0.5 µg of bovine serum albumin or 8 µg of pooled seminal plasma (normozoospermic group n = 10) for SyproRuby staining and lectin blotting respectively. SERVA Triple Color Protein Standard III (SERVA Electrophoresis GmbH, Heidelberg, Germany) was used for calculation of glycoprotein molar masses. Gels and blots were acquired with Gbox F2 CCD camera (Syngene, Cambridge, UK).

### 4.3. Lectin-Blotting

*Maclura pomifera* and *Vicia Villosa* lectins (Vector Labs, Peterborough, UK) were applied to detect T or Tn antigens in SP glycoproteins. After overnight blocking with 1% Tween-20 solution, the membranes were incubated with biotinylated lectins diluted 1:1000 for MPL or 1:2000 for VVL in 15 mM Tris–HCl buffer, pH 7.4, with 0.15 M NaCl and 0.1% Tween-20 (TBST) for 1 h at room temperature (RT), with constant gentle shaking. The glycoprotein-lectin complexes were detected with ExtrAvidin-alkaline phosphatase (Sigma–Aldrich, St. Louis, MO, USA, diluted 1:10,000 in TBST, 45 min, RT) and BCIP/NBT substrate (Sigma-Aldrich, St. Louis, MO, USA). Glycophorin A isolated from human erythrocytes with ‘mucin-like’ O-glycosylation pattern was used as a positive control for lectin reactivity, and yeast carboxipeptidase Y (Sigma–Aldrich, St. Louis, MO, USA) as a negative control.

### 4.4. Densitometric Analysis

GeneTools 4.03 (Syngene, Cambridge, UK) software [84] was used for the semi-quantitative analysis of gels and blots. Lectin reactivity was expressed in arbitrary units (AU) defined as the ratio of optical density (OD) of the MPL- or VVL-stained band of interest over OD of the corresponding band of pooled seminal plasma loaded on each gel in constant protein amount (ODx/ODxref).

To examine whether the possible alterations in the lectin reactivity are related to the changed protein amount or to the intensity of the glycosylation process itself, the density of glycoepitopes of interest was calculated as the ratio of MPL or VVL reactivity to protein content in each individual electrophoretic band. The latter was estimated as the ratio of sample/BSA band signals from SyproRuby stained gels (OD_x_/OD_BSA_) [78].

To verify the lectin reactivity ratio of VVL over MPL for the 70kDa band (to estimate Tn/T ratio), the standardized OD of VVL-stained band was divided by the standardized OD of MPL-stained band for each sample, respectively.

### 4.5. Affinity Chromatography

MPL-reactive or VVL-reactive glycoproteins were isolated by means of affinity chromatography. Lectins were immobilized onto NHS-activated agarose resin (Thermo Fisher Scientific, Waltham, MA, USA) in the spin columns in the proportion of 2 mg of lectin per 1 mL of swelled resin. The coupling reaction was carried out for 2 h at room temperature with gentle stirring, in PBS buffer, pH = 7.5. To block the remaining NHS-activated group, columns were incubated with 1 M Tris-HCl pH-7.4 for 15 min with end-over-end stirring. A seminal plasma sample was prepared by pooling the equal protein amount of 10 normozoospermic subjects. Bulk seminal plasma sample was diluted in PBS and an aliquot of the sample containing 1 mg of protein was loaded on each column. The ligand-lectin binding reaction was carried out overnight at 4 °C with agitation. The unbound proteins were washed out of the column and MPL-reactive or VVL-reactive glycoproteins were eluted with Gly-HCl buffer, pH = 3.0 or 0.2 M GalNAc in 0.2 M acetate buffer, pH = 4.0. The released proteins were immediately neutralized, buffer exchanged against PBS, washed three times and concentrated on protein concentrators (Thermo Fisher Scientific, Waltham, MA, USA) MWCO 3 kDa, centrifuged three times for 30 min with 10,000× *g*, adding 400 µL PBS each time). 10 μL of each isolate was separated by SDS-PAGE in a 12.5% gel and to detect proteins, silver-staining procedure was applied [85].

### 4.6. Liquid Chromatography–Mass Spectrometry (LC-MS) Identification of Glycoproteins

Proteomic profiling of lectin-reactive SDS-PAGE bands and fractions eluted from the affinity columns were carried out by mass spectrometry analysis performed in Laboratory of Mass Spectrometry, Institute of Biochemistry and Biophysics, Polish Academy of Sciences, Warsaw, Poland. After SDS-PAGE of pooled SP samples (n = 10), selected bands were cut off the CBB (Coomassie Brilliant Blue) stained gel (0.5% CBB in 40% methanol and 10% acetic acid). Trypsin digestion was performed with the standard Shevchenko procedure [86]. For in-solution trypsin digestion, the samples (column isolates) were reduced with dithiothreitol (10 mM) in 100 mM NH_4_HCO_3_ (30 min, 57 °C), and alkylated with 50 mM iodoacetamide (IAA, 45 min in the dark, RT). Trypsin was added in the enzyme:protein ratio 10:1, then the sample was digested overnight at 37 °C, followed by acidification with 5% trifluoroacetic acid (TFA). Mass spectrometric analyses were done with a Q Exactive Focus Hybrid Quadrupole-Orbitrap Mass Spectrometer (Thermo Fisher Scientific, Waltham, MA, USA) and searched with the Mascot engine (Matrix Science, Boston, MA, USA) through the Swiss Prot database (Expasy, The Swiss Bioinformatics Resource Portal, Swiss Institute of Bioinformatics, Lausanne, Switzerland, access: 13 October 2020, 11 December 2020). Trypsin was specified as the proteolytic enzyme and 1 missing cleavage was allowed. Carbamidomethylation of cysteine was set as fixed modification and methionine oxidation was set as the only variable modification. Charge state of +1 was considered for parent ions. Tolerance of ±30 ppm was set for a peptide and ±0.1 Da for MS/MS fragment mass accuracy. Only peptides identified with a significant Mascot score (*p* < 0.05) were considered.

### 4.7. Reactivity with Antibodies

To confirm the presence of individual proteins in the isolates, proteins eluted with Gly-HCl buffer and 0.2 M GalNAc in acetate buffer were pooled, separated by SDS-PAGE and transferred onto a nitrocellulose membrane. Blots were incubated with the following antibodies: polyclonal rabbit anti-semenogelin 1 (1:1000, MBS968887, MyBioSource, San Diego, CA, USA), anti-semenogelin 2 (1:1000, MBS9132638, MyBioSource, San Diego, CA, USA) anti-lactotransferrin (1:1000, ab197811, Abcam, Cambridge, UK), anti-prolactin-inducible protein (1:600, ab198018, Abcam, Cambridge, UK), anti-fibronectin (1:2000, F3648, Sigma–Aldrich, St. Louis, MO, USA) and next with horseradish peroxidase labeled goat anti-rabbit IgG (1:5000, 111-035-045, Jackson Antibodies, Suffolk, UK). For HRP staining diaminobenzidine (0.1 mg/mL) and H_2_O_2_ (0.04%) in 0.1 M citric buffer, pH 5.5, were applied.

### 4.8. Bioinformatics Analysis

O-glycosylation potential in selected proteins was analyzed using NetOGlyc 4.0 Server (DTU Health Tech, Lyngby, Denmark, access 15 February 2020–10 March 2020) [87].

The STRING 11.0 database (STRING Consortium, Swiss Institute of Bioinformatics, Lausanne, Switzerland, access 5 March 2021) was used to identify predicted functional partners and protein–protein interaction networks (PPI) of the most abundant proteins, identified in the selected SDS-PAGE bands and proteins comprised in chromatographic lectin-reactive fractions. Analysis was performed using the evidence mode involving seven types of evidence (neighbourhood in the genome, gene fusions, co-occurrence across genomes, co-expression and experimental/biochemical data associated with curated databases, co-mentioned in Pubmed abstracts). PPI network included no more than 5 of interactions of 1st shell with the medium confidence (0.400). Functional enrichment analysis was performed for gene ontology and reactome pathway using the whole genome as a background. 

### 4.9. Statistical Analysis

Statistica 13.0 software (StatSoft Polska, Kraków, Poland [88]) was used for statistical analysis. Box–whisker graphs with median values and the inner quartiles of reactivity range were used to show the results of lectin reactivity and density of particular glycoepitopes. Statistical significance of the difference among the groups was analyzed with the Kruskal–Wallis test, with *p* < 0.05 considered to be significant.

## 5. Conclusions

To conclude, we have identified for the first time, seminal plasma proteins bearing GalNAc and Gal terminated glycans, and potential ligands for endogenous Gal/GalNAc specific lectins, which may participate in the modulation of maternal immune response during fertilization.

## Figures and Tables

**Figure 1 ijms-22-07331-f001:**
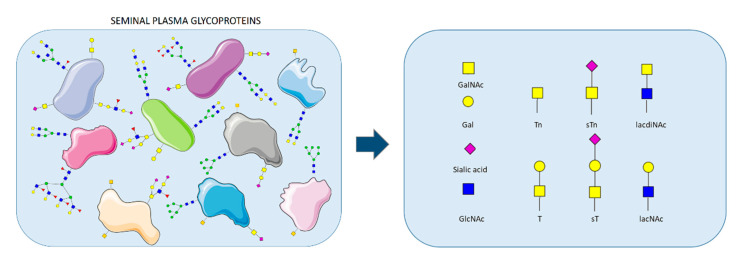
The schematic representation of seminal plasma glycan structures, highlighting the antigens analyzed in this study. Images were taken from Servier Medical Art (https://smart.servier.com (accessed on 14 June 2021)) and modified by the authors under the following terms: Creative Commons Attribution 3.0 Unported License.

**Figure 2 ijms-22-07331-f002:**
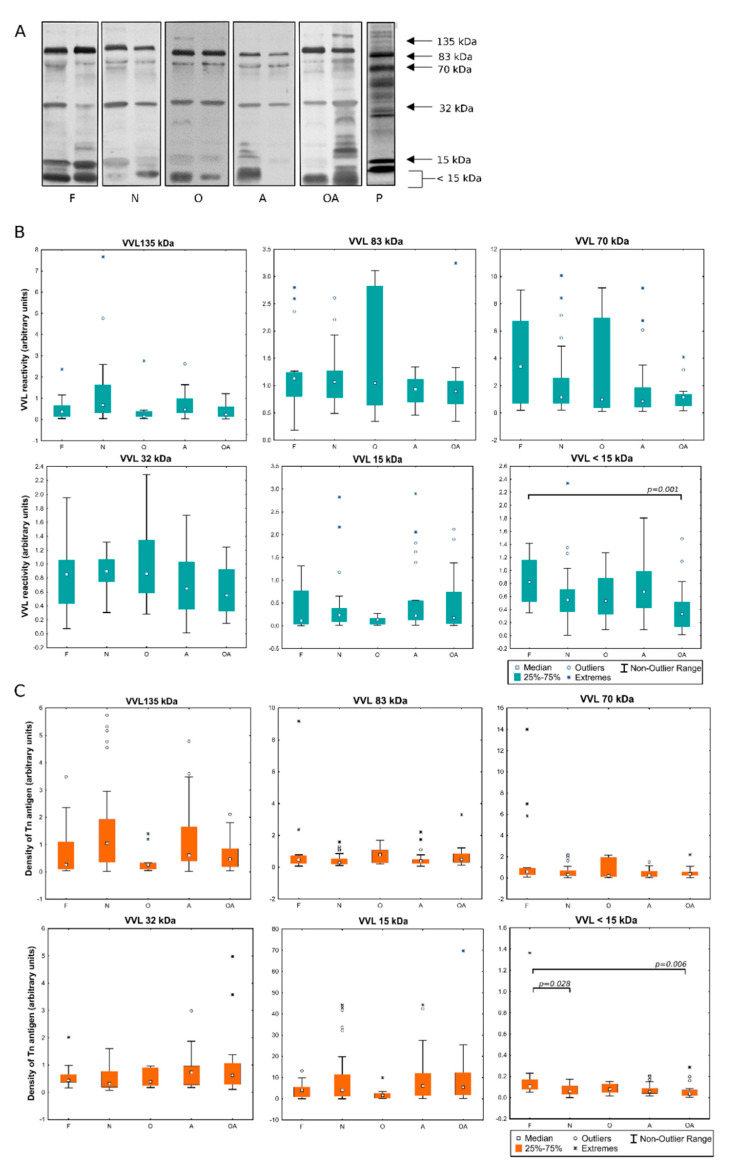
*Vicia villosa* reactivity of seminal plasma. (**A**)—representative lectin-blotting patterns of individual samples classified to different diagnostic groups: F: normozoospermic fertile (control); N: normozoospermic non-fertile; O: oligozoospermic; A: asthenozoospermic; OA: oligoasthenozoospermic; P: SDS-PAGE seminal plasma protein bands stained with SyproRuby (negative); (**B**)—total reactivity in the selected bands measured as densitometric analysis; (**C**)—density of VVL-detected glycans calculated as VVL/protein ratio.

**Figure 3 ijms-22-07331-f003:**
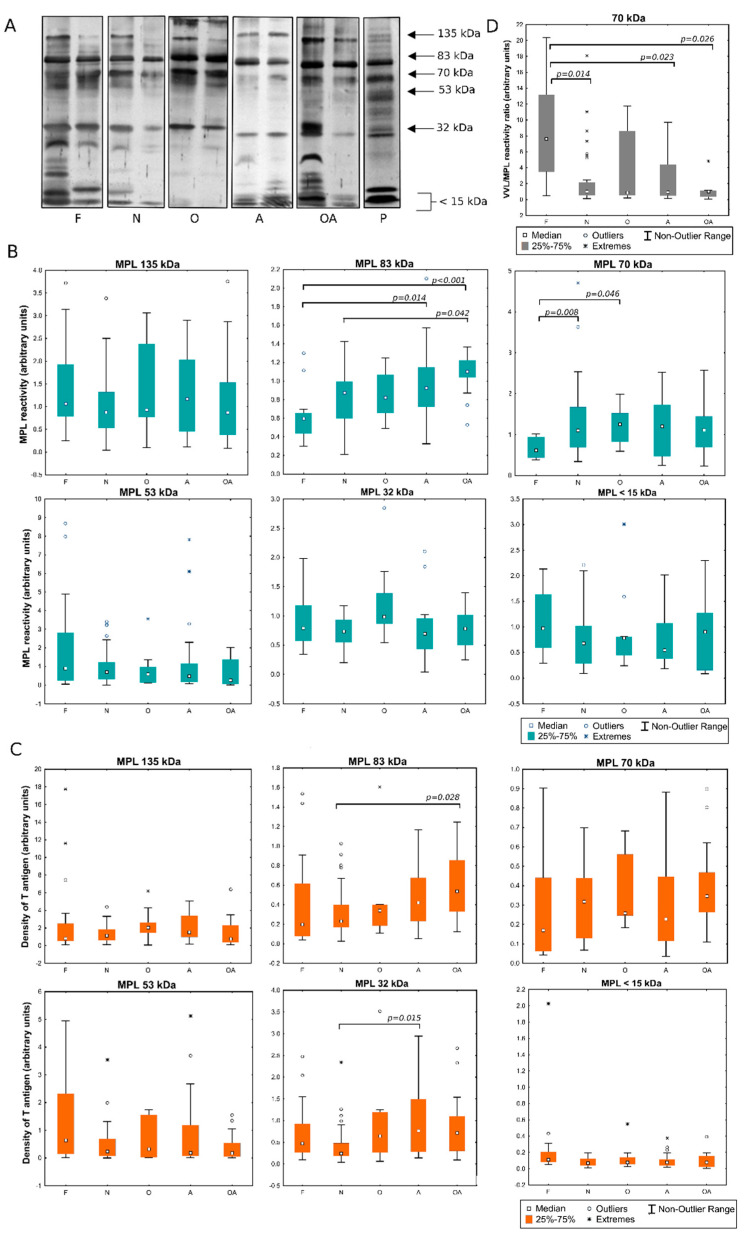
*Maclura pomifera* reactivity of seminal plasma. (**A**)—representative lectin-blotting patterns of individual samples classified to different diagnostic groups: F: normozoospermic fertile (control); N: normozoospermic non-fertile; O: oligozoospermic; A: asthenozoospermic; OA: oligoasthenozoospermic; P: SDS-PAGE seminal plasma protein bands stained with SyproRuby (negative); (**B**)—total reactivity in the selected bands measured as densitometric analysis; (**C**)—density of MPL-detected glycans calculated as MPL/protein ratio; (**D**)—lectin reactivity ratio of VVL to MPL for the 70 kDa band.

**Figure 4 ijms-22-07331-f004:**
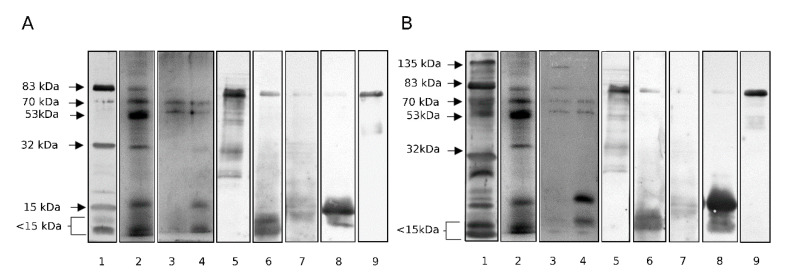
Protein and immunoreactivity patterns of the lectin affinity chromatography isolates. Lane 1: lectin probed whole seminal plasma; 2: silver stained whole seminal plasma; 3: 0.2 M GalNAc released glycoproteins, silver stained; 4: Gly-HCl buffer released glycoproteins, silver stained; Lanes 5–9: immunoreactivity of the isolates; 5: a-FN; 6: a-SEMG1; 7: a-SEMG2; 8: a-PIP; 9: a-LTF. (**A**)—VVL-agarose column, (**B**)—MPL-agarose column.

**Figure 5 ijms-22-07331-f005:**
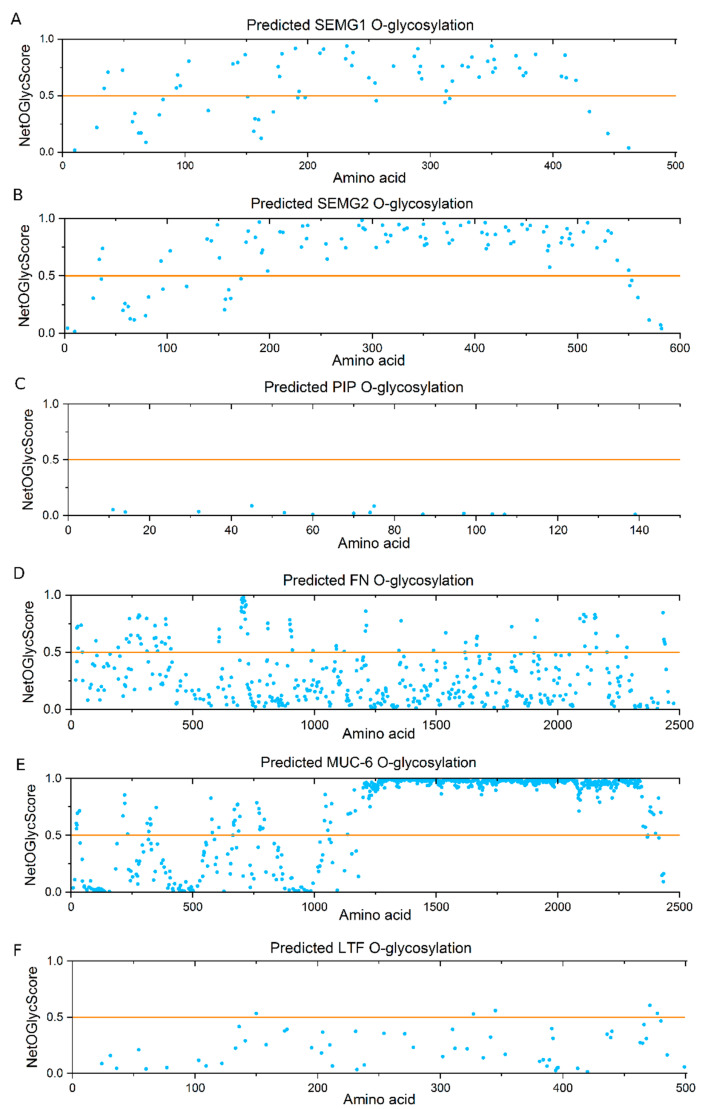
NetOGlyc analysis of the potential O-glycosylation sites on Ser/Thr residues in the selected seminal plasma glycoproteins. Residues with NetOGlyc 4.0 scores >0.5 indicate predicted O-linked glycans. Glycoproteins analyzed: (**A**)—SEMG1; (**B**)—SEMG2; (**C**)—PIP; (**D**)—FN; (**E**)—MUC6; (**F**)—LTF.

**Figure 6 ijms-22-07331-f006:**
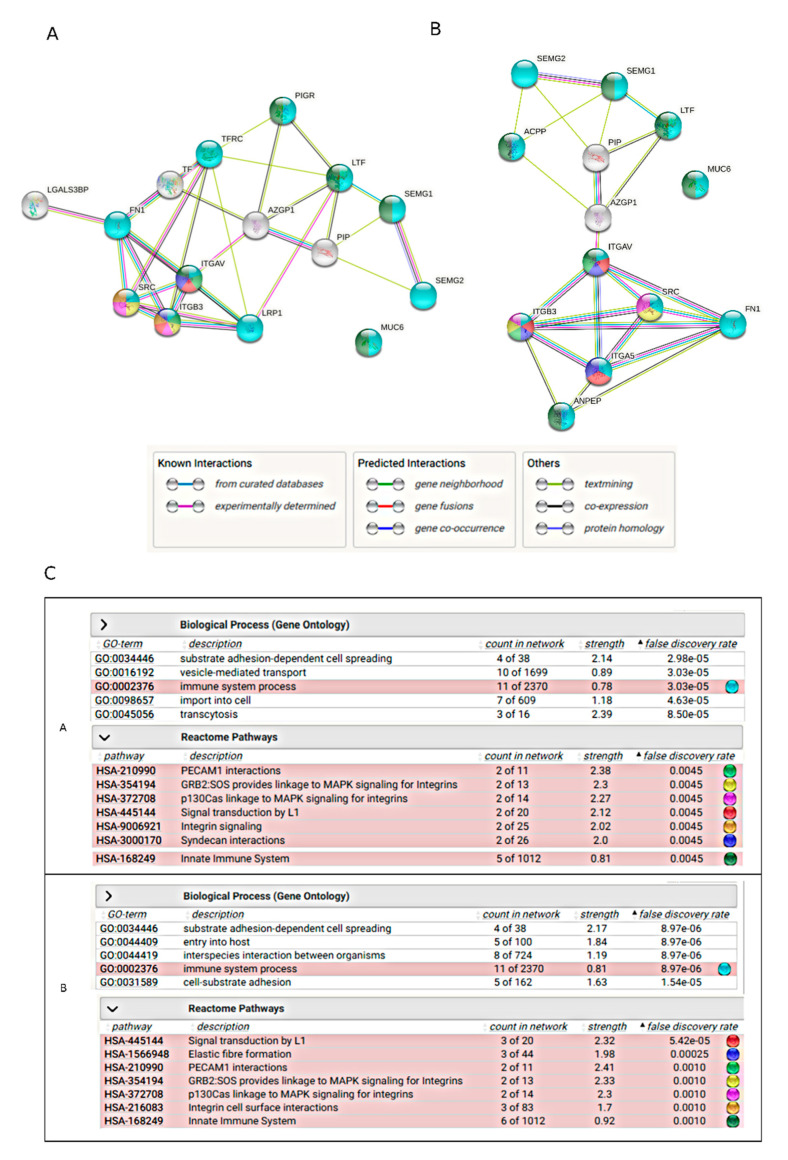
STRING analysis of the networks of glycoproteins identified as carriers of Gal and GalNAc epitopes. (**A**)—glycoproteins identified in SDS-PAGE bands; (**B**)—glycoproteins identified in affinity chromatography isolates, (**C**) —STRING report of the functional enrichment analysis for gene ontology and reactome pathways.

**Table 1 ijms-22-07331-t001:** LC-MS protein identification of lectin-reactive bands separated by SDS-PAGE.

Fraction	Protein	Score	Matches	Molar Mass [Da]
135 kDa	Mucin 6	12,688	271	263,159
	Aminopeptidase N	8732	204	109,870
	Serum albumin	5456	111	71,317
	Fibronectin	5109	111	266,052
	Lactotransferrin	4699	96	80,014
	Dipeptidyl peptidase 4	3143	75	88,907
	Laminin subunit beta 2	2410	56	202,982
	Beta mannosidase	1174	31	101,800
83 kDa	Lactotranferrin	43,323	872	83,000
	Fibronectin	14,382	309	266,052
	Serum albumin	4267	108	71,317
	Serotransferrin	2896	78	79,294
	Mucin-6	2692	62	263,159
	Polymeric immunoglobulin receptor	1972	37	84,429
	Alpha 2 macroglubulin	1049	32	164,613
70 kDa	Serum albumin	38,789	946	71,317
	Galectin 3 binding protein	2960	59	66,202
	Lactotransferrin	1668	44	80,014
	Fibronectin	1061	41	266,052
53 kDa	Prostatic acid phosphatase	12,624	443	44,880
	Alpha 1 antitrypsin	3616	139	46,878
	Fibronectin	1528	46	266,052
	Lactotransferrin	1390	42	80,014
32 kDa	Fibronectin	9291	221	266,052
	Prostate specific antigen	6171	222	29,293
	Annexin	4182	80	35,971
	Clusterin	2358	66	53,031
	Prostatic acid phosphatase	1381	35	44,880
	Prosaposin	1263	42	59,899
	Serum albumin	1098	42	71,317
	Carboxypeptidase E	1059	34	53,516
	Cysteine rich secretory protein 1	1026	19	29,432
	Purine nucleoside phosphorylase	1010	23	32,325
15 kDa	Prolactin inducible protein	37,959	916	16,847
	Peptidyl prolyl cis trans isomerase A	1145	33	18,229
	Prostatic acid phosphatase	1084	37	44,880
<15 kDa	Prolactin inducible protein precursor	1823	39	16,847
	Semenogelin 1 preproprotein	1134	27	52,157
	Prolactin induced protein	1090	23	9232
	Semenogelin 2 precursor	1023	21	65,519

**Table 2 ijms-22-07331-t002:** LC-MS identification of proteins isolated in lectin affinity chromatography.

VVL			
Protein	Score	Matches	Molar mass (Da)
Semenogelin-2	2997	135	65,497
Prolactin-inducible protein	1482	39	16,792
Semenogelin-1	1278	49	52,146
Lactotransferrin	504	16	79,650
Prostate-specific antigen	86	2	29,183
Fibronectin	67	2	27,5047
**MPL**			
Protein	Score	Matches	Molar mass (Da)
Prolactin-inducible protein	9346	158	16,792
Semenogelin-2	6473	159	65,497
Semenogelin-1	3895	80	52,146
Fibronectin	1471	28	275,047
Lactotransferrin	1440	29	79,650
Prostate-specific antigen	809	21	29,183
Aminopeptidase N	758	16	109,793
Prostatic acid phosphatase	539	11	44,813
Clusterin	242	6	52,921
Zinc-alpha-2-glycoprotein	179	3	34,421
Mucin-6	117	2	261,946
Glycodelin	97	2	29,244

**Table 3 ijms-22-07331-t003:** Detailed characteristics of patients’ age.

Group	Age (Years)		
	Mean ± SD	Median	Mode
F	34 ± 2.73	33	32
N	33 ± 3.69	32	32
O	35 ± 6.13	33	33
A	34 ± 4.72	32.5	32
OA	32 ± 3.15	31	30

Seminal plasma groups: F: normozoospermic fertile (control); N: normozoospermic non-fertile; O: oligozoospermic; A: asthenozoospermic; OA: oligoasthenozoospermic.

## Data Availability

The data underlying this article will be shared on reasonable request to the corresponding author.

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
