# Peer review of "Glycoproteins Presenting Galactose and N-Acetylgalactosamine in Human Seminal Plasma as Potential Players Involved in Immune Modulation in the Fertilization Process"

_ijms, 2021, doi:10.3390/ijms22147331_

Round 1
Reviewer 1 Report
This is an interesting and well-conducted research on the presence of glycoproteins in human seminal plasma implicated in the modulation of maternal immuno response. The manuscript is clear and the results obtained are very interesting and exposed in an orderly way.
I only have few minor comments:
1) Please, use the past tense in the Abstract section.
2) Linees 39-40. "Numerous more recent data" is not a suitable expression. Please, change it.
3) Authors must revise the whole document and correct minor spelling and grammatical errors.
Author Response
The manuscript has been revised according to the Reviewer’s remarks
1) Please, use the past tense in the Abstract section.
RESPONSE 1 - the past tense was used in the abstract section
2) Linees 39-40. "Numerous more recent data" is not a suitable expression. Please, change it.
RESPONSE 2 – it was change to “some recent data”
3) Authors must revise the whole document and correct minor spelling and grammatical errors.
RESPONSE 3 – The manuscript has been reviewed by English language professional spelling and grammar errors were corrected and highlighted in the text
Reviewer 2 Report
Research article entitled „Galactose and N-acetyl-galactosamine in Human Seminal Plasma as Potential Players Involved in Modulation of Maternal Immune Response During Fertilization Process” received in IJMS raise a very interestig point of view on selected glycoproteins in mechanism of immunosupression of female respons on semen. An advanced analysis was carried out and obtained results showed many differences between analysed samples. What indicate that the type of disfunction play a major role in occurence various types and amount of examined glycoproteins.
However, in my opinion, it would be good to introduce an additional division by age of patients because as we know, the quality of semen changes drastically with age. Result of 22 year old man and result from 56 year old man may vary because thay are on two different stage of physiological status (i.e testosteron level or efficiency of aromatisation in testis), especially if they have similar sperm disfunction. Thus my proposition is add more physiological context to discussion.
I would also suggest changing the title of the manuscript, as the current title suggests that the research was also conducted on the female reproductive system and this may be confusing to the reader.
Generally I have no more objection to this research paper.
Author Response
Replay to Rev.2
Remark:
However, in my opinion, it would be good to introduce an additional division by age of patients because as we know, the quality of semen changes drastically with age. Result of 22 year old man and result from 56 year old man may vary because thay are on two different stage of physiological status (i.e testosteron level or efficiency of aromatisation in testis), especially if they have similar sperm disfunction. Thus my proposition is add more physiological context to discussion.
Response 1
We agree that the description of the samples is insufficient and may lead to some doubts. Actually only two samples originated from men aged ³50 (50 and 56). To avoid misunderstanding in this aspect we have prepared more detailed data, provided in a form of table comprising mean±SD, median and mode value. Regarding these data we suppose that the impact of possible physiological difference reflecting the patients’ age is not significant. As the biological material is hardly available, we considered each single sample valuable. Anyway, this objection is important for us, and limitation of the age range will be taken into account in our future research.
Remark:
I would also suggest changing the title of the manuscript, as the current title suggests that the research was also conducted on the female reproductive system and this may be confusing to the reader.
Response 2
The title was shortened to avoid confusion
We highly appreciate the reviewer’s time and efforts devoted to evaluate our article and hope our response is satisfactory.